# Comparison of rainfall generators with regionalisation for the estimation of rainfall erosivity at ungauged sites

Ross Pidoto[1], Nejc Bezak[2], Hannes Müller-Thomy[3,4,+], Bora Shehu[1], Ana Claudia Callau-Beyer[5], Katarina Zabret[2], Uwe Haberlandt[1]

[1]Institute of Hydrology and Water Resources Management, Leibniz University Hannover, Germany
[2]University of Ljubljana, Faculty of Civil and Geodetic Engineering, Ljubljana, Slovenia
[3]Leichtweiß Institute for Hydraulic Engineering and Water Resources, Department of Hydrology, Water Management and Water Protection, Technische Universität Braunschweig, Brunswick, Germany
[4]Institute of Hydraulic Engineering and Water Resources Management, Vienna University of Technology, Austria
[5]Institute of Horticultural Production Systems, Leibniz University Hannover, Germany
+previously published under the name Hannes Müller

*Correspondence to*: Hannes Müller-Thomy (h.mueller-thomy@tu-braunschweig.de)

**Abstract.** Rainfall erosivity values are required for soil erosion prediction. To calculate the mean annual rainfall erosivity (R), long-term high-resolution observed rainfall data is required, which is often not available. To overcome the issue of limited data availability in space and time, four methods were employed and evaluated: direct regionalisation of R, regionalisation of 5 min rainfall, disaggregation of daily rainfall into 5 min timesteps, and a regionalised stochastic rainfall model. The impact of station density is considered for each of the methods. The study is carried out using 159 recording and 150 non-recording (daily) rainfall stations in and around the federal state of Lower Saxony, Germany. In addition, the minimum record length necessary to adequately estimate R was investigated. Results show that the direct regionalisation of mean annual erosivity is best in terms of both relative bias and relative root mean square error (RMSE), followed by the regionalisation of the 5 min rainfall data, which yields better results than the rainfall generation models, namely an alternating renewal model (ARM) and a multiplicative cascade model. However, a key advantage of using regionalised rainfall models is the ability to generate time series that can be used for the estimation of the erosive event characteristics. This is not possible if regionalising only R. Using the stochastic ARM, it was assessed that more than 60 years of data is needed in most cases to reach a stable estimate of annual rainfall erosivity. Moreover, the temporal resolution of measuring devices was found to have a significant effect on R, with coarser temporal resolution leading to a higher relative bias.

## 1 Introduction

Intense soil erosion has a significant impact on the environment, for example presenting a major threat for agricultural production or leading to increased sedimentation and pollution in rivers, which also affects aquatic organisms. Soil erosion modelling can be performed to detect critical parts of the landscape and design suitable counter measures to reduce soil losses. One of the most frequently applied models for soil erosion modelling is the Universal Soil Loss Equation (USLE) and its successor the Revised Universal Soil Loss Equation (RUSLE) (e.g., Renard et al., 1997). In the scope of the RUSLE model, soil erosion is described by six factors, one of which is the rainfall erosivity factor R (in MJ mm ha$^{-1}$ h$^{-1}$, Renard et al., 1997). Rainfall erosivity is often characterised by large spatial and temporal variability (e.g., Bezak et al., 2020, 2021a, 2021b; 2021c; Panagos et al., 2016, 2017; Petek et al., 2018), meaning that  its estimation is not straightforward and requires adequate temporal and spatial data resolution.

In order to obtain robust rainfall erosivity values, high-resolution observed rainfall data are needed, in time ideally with a 1- or 5-minute resolution (Dunkerley, 2019), and in space ideally to represent the spatial heterogeneity (Peleg et al., 2021). However, rainfall data at this resolution is often only available for shorter periods of observation (e.g. 10 or 20 years) or not at all. A solution to overcome this shortcoming is the use of stochastic rainfall models that allow the generation of long time series of arbitrary length. Additionally, these models can generate time series for ungauged locations through regionalisation of model parameters. In some cases, gridded rainfall erosivity data is used as input to the USLE-type soil erosion models. In terms of spatial description of the rainfall erosivity most often station-based data is interpolated (e.g., Panagos et al., 2017) while satellite-based data could in the future provide useful estimates of gridded rainfall erosivity (e.g., Bezak et al., 2022). However, due to the station-based approach of the USLE, the generation of spatial rainfall is not required, but would be useful for more sophisticated approaches to estimate erosion (Eekhout et al., 2021). A few studies have investigated the possibility of applying stochastic precipitation models to generate rainfall time series to then estimate rainfall erosivity (e.g., Jebari et al., 2012; Lobo et al., 2015; de Oliveira et al., 2018; Haas et al., 2018). Methods used to model rainfall include cluster-based models (e.g., Onof et al., 2000, Onof and Wang, 2020), cascade models (e.g., Molnar and Burlando, 2005; Pohle et al., 2018, Müller and Haberlandt, 2018), method-of-fragments (e.g., Breinl and Di Baldassarre, 2019) or alternating renewal models (e.g., Callau Poduje and Haberlandt, 2017), or as part of weather generators (Peleg et al., 2017). The parameters of these methods are estimated based on observations and the complexity of the models generally depends on the target temporal resolution. Thus, most studies are focused on daily data. For example, CLImate GENerator (CLIGEN) was applied to obtain daily rainfall estimates and calculate rainfall erosivity using daily data (e.g., de Oliveira et al., 2018; Lobo et al., 2015; Wang et al., 2018). Shortcomings such as sensitivity to the input parameters have been reported in the literature (e.g., Meyer et al., 2008; Haas et al., 2018). On the other hand, temporal high-resolution time series (i.e. 5 minutes) are less often generated using stochastic rainfall models, although in recent years advancements have been made (e.g., Haberlandt et al., 2008; Vandenberghe et al., 2011; Vernieuwe et al., 2015, Callau Poduje and Haberlandt, 2017, Müller-Thomy, 2020).

Thus, a few studies (e.g., De Oliveira et al., 2018; Haas et al., 2018) have investigated if stochastic rainfall models are able to correctly predict rainfall erosivity patterns at specific locations. In the case that a stochastic rainfall model is able to mimic the rainfall erosivity characteristics, generated long-term high-resolution rainfall time series should then allow a robust estimation

of annual and even monthly erosivity patterns. Similarly, a limited number of studies (e.g., Angulo-Martinez et al., 2009) have investigated performance of different interpolation techniques related to the mapping of rainfall erosivity.

The main aim of this study is to evaluate and compare different rainfall generators and regionalisation approaches in order to obtain either directly or indirectly annual rainfall erosivity estimates for ungauged locations. Given a lack of high-resolution rainfall time series, the research question is whether these tested methods can adequately reproduce observed annual rainfall.

As a follow-up research question, we investigate the performance of tested methods in terms of specific erosive events characteristics. This information is not directly needed as input to the USLE-type models but is often studied and investigated in rainfall erosivity studies. Finally, given the existence of a high-resolution rainfall time series, we also investigated how long should these time series be in order to obtain stable site annual rainfall erosivity? Hence, this information is relevant both for the soil erosion model applications using USLE-type models as well as for the studies investigating erosive event

characteristics.

All tests were performed via leave-one-out cross validation, as the premise of this study is that high-resolution time series are not widely available. Additionally, the effect of station density on the regionalisation performance was assessed by performing each test with five different station counts (20%, 40%, 60%, 80% and 100% of observed stations). To minimise the sampling uncertainty, 20 realisations of each test at each station density were performed.

**2 Data and study area**

High-resolution observed rainfall data for 159 stations bounded by the rectangle 7°E to 12°E and 51°N to 54°N (Fig. 1), centred over the German federal state of Lower Saxony, was acquired from the German Weather Service (DWD). The one-minute source time series, from a combination of tipping bucket and later drop counter measurements, were aggregated to 5 minutes for use within this project. All data used for the project is restricted to the ten-year period 2007-2016 to maximise the data

availability across all stations, as all 159 stations have at least 98% data availability for this period. The study area is dominated by lowland terrain, with the only mountains of significance being the southeast lying Harz mountain range. The region is predominantly classified as having a temperate oceanic (Cfb) climate according to Köppen-Geiger (Kottek et al., 2006). The far eastern portion of the study area is categorised as temperate continental (Dfb) and the Harz mountain range as cool continental (Dfc). Fig. 2 displays the long-term seasonal rainfall and temperature variability averaged over the German federal

state of Lower Saxony. Annual rainfall varies between 950 mm in the Harz mountain range to under 400 mm in lower lying areas east of the study area. As can be seen from Fig. 1 and Fig. 2, there is for most locations no significant difference between summer and winter precipitation in the study area. In terms of monthly rainfall erosivity, higher values (i.e. around 100 $MJ.mm.ha^{-1}.h^{-1}.month^{-1}$) can be seen for summer followed by spring and autumn, whilst the lowest rainfall erosivity values

could be seen for winter (i.e. less than 20-30 MJ.mm.ha$^{-1}$.h$^{-1}$.month$^{-1}$). The results from a study conducted by Ballabio et al.

(2017) are in accordance to the results of this study.

## 3 Methods

In this study, four different methods to calculate annual rainfall erosivity (R) for ungauged sites were applied (Fig. 3). The first and simplest method is the direct regionalisation of observed mean annual rainfall erosivity (method Direct-R). The other three

methods all generate rainfall time series first, from which mean annual rainfall erosivity is subsequently calculated. The second method is the regionalisation of the observed 5-minute rainfall time series (method Direct-P). The final two methods are stochastic rainfall models – one being a regionalised disaggregation model using regionalised daily rainfall as its source (method Disagg) (Müller-Thomy, 2020) and the other a regionalised alternating renewal model (method ARM) (Callau Poduje and Haberlandt, 2017). Detailed descriptions of the applied methods are provided in sections 3.2 and 3.3 while section 3.1

provides details about the rainfall erosivity calculation.

### 3.1 Rainfall erosivity

Rainfall erosivity is one of the factors that has the highest impact on soil erosion rates. Rainfall erosivity is characterised by multiple properties of rainfall events such as the kinetic energy of raindrops, rainfall intensity and rainfall duration. In order to calculate annual rainfall erosivity for a selected time span, the following equation proposed by Renard et al. (1997) can be

used:

$$R = \frac{\sum_n E \cdot I_{30}}{M} \qquad (1)$$

where $R$ is rainfall erosivity (MJ mm ha$^{-1}$ h$^{-1}$), $I_{30}$ the maximum 30-minute rainfall intensity for a specific rainfall event (mm h$^{-1}$), $E$ the kinetic energy of the rainfall event (MJ ha$^{-1}$ ), $n$ the total number of erosive events and $M$ the time span (yrs). Equation 2 is used to derive the kinetic energy of the rainfall event:

$$E = e_B \cdot I \cdot \Delta t \qquad (2)$$

where $e_B$ is the specific kinetic energy (MJ ha$^{-1}$ mm$^{-1}$), $I$ the rainfall intensity (mm h$^{-1}$) for the event and $\Delta t$ the time interval (h). Various equations for calculation of $e_B$ were developed for different parts of the world, where most often high frequency measurements using optical disdrometers are used to derive the $e_B – E$ relationship (e.g., Petan et al., 2010). One of the most frequently and globally used equations (e.g., Panagos et al., 2015, 2017) was proposed by Brown and Foster (1987):

$$e_B = 0.29 \cdot [1 - 0.72 \cdot exp\,(-0.05 \cdot I)\,] \qquad (3)$$

This equation is also mentioned in the RUSLE Handbook (Renard et al., 1997), and is regarded as the standard method to estimate soil erosion using the RUSLE approach. Additionally, the equation has been already applied several times to similar

climatic conditions (e.g., Panagos et al., 2015) and yielding meaningful results. This equation was used in the present study in order to calculate specific kinetic energy. Equation 3 is valid for rainfall intensities ranging between 0 and 250 mm/h.

Erosive rainfall events are defined according to the RUSLE methodology (Renard et al., 1997). A rainfall event is considered erosive if the total volume exceeds 12.7 mm of rain or if the maximum volume in 15 minutes is more than 6.35 mm. Here a 6-hour period without rain is used in order to separate two consecutive erosive rainfall events. The monthly and annual rainfall erosivity values are then calculated based on the rainfall erosivity of all erosive events.

## 3.2 Direct regionalisation methods

**3.2.1 Method Direct-R: direct regionalisation of mean annual rainfall erosivity R**

The direct interpolation of erosivity was carried out using geostatistical methods (see e.g. textbooks from Isaaks and Srivastava, 1990 or Goovaerts, 1997). The spatial dependence of $R$ is modelled using an isotropic spherical variogram model:

$$\gamma(h) = c_0 + c \left( \frac{3}{2} \frac{h}{a} - \frac{1}{2} \frac{h^3}{a} \right) if\ h \le a,\ otherwise\ \gamma(h) = c_0 + c \tag{4}$$

where $c$ is 0.5, $c_0$ is 0.3 and $a$ is 30000 m. For interpolation, ordinary kriging (OK) and external drift kriging (EDK) are applied.
While OK assumes spatial stationarity, EDK relaxes the intrinsic hypothesis regarding the constant mean E and assumes a linear relationship between the mean of the target variable Z and one or more additional variables Y:

$$E[Z(u)|Y(u)] = a + \sum_{i=1}^{n} b_i Y_i(u) \tag{5}$$

The coefficients a and $b_i$ need not be known explicitly, but are considered in building the kriging system. The additional variables $Y_i$ need to be provided for all points with observed erosivity and for all unobserved points to be interpolated, usually
on a raster. Here, the following additional variables are used in the analyses: daily mean rainfall (*Pmean*), maximum daily rainfall (*Pmax*), the 90 and 99 percent quantiles of daily rainfall (*Pq90*, *Pq99*) and the yearly wet fractions of days with rainfall higher than 12.7 mm (based on the RUSLE methodology; Renard et al., 1997) and 1 mm thresholds (*wfd12*, *wfd1*). These variables are extracted from regionalised daily data in a spatial resolution of 1 km² (NLWKN, 2019), and are calculated as long-term averages from the period 2007-2016. The interpolation uses a minimum of 8 neighbours and a maximum of 12
neighbours within a radius of 300 km. The R statistical software (R core team, 2015) with the package "gstat" (Pebesma, 2004) is employed for the calculations. It should be noted that some other variables could have been selected such as mean annual precipitation, which is often used to estimate the mean annual rainfall erosivity in data sparse regions. However, since this study uses high-resolution data, we focused on specific precipitation characteristics that could be obtained from these high-resolution data.

### 3.2.2 Method Direct-P: regionalisation of 5-minute rainfall

A second approach for the regionalisation of the annual R is to estimate the 5-minute rainfall time series for unobserved locations based on the information of nearby measured rainfall time series. In this study, the nearest neighbour approach (NN) was used for the regionalisation of the 5-minute rainfall time series from 2007-2016. This technique, also known as the Thiessen Polygon method (Thiessen, 1911), is a simple approach which assigns for each unobserved point the observed rainfall time series of the closest available rain gauge. Despite this being a basic interpolation method, a prior investigation of different interpolation techniques (nearest neighbour, inverse distance and ordinary kriging) led to the conclusion that this technique maintains the best temporal structure of rainfall at the 5-minute time scale and yields the lowest error for the calculation of erosivity. One reason for the superiority of NN compared to the other interpolation methods might be that NN is the only method which does not smooth extremes, which is important for the estimation of erosive events.

### 3.3 Stochastic rainfall models

#### 3.3.1 Method ARM: an alternating renewal rainfall model

Stochastic rainfall models allow for the generation of rainfall time series of arbitrary length, including for unobserved locations through regionalisation. For this study, the alternating renewal model (ARM) based on the theory of renewal processes was used to generate 5-minute synthetic rainfall time series. In this model, rainfall is described as a series of independent alternating wet and dry spells, described by the three variables: wet spell amount (WSA), wet spell duration (WSD) and dry spell duration (DSD) (Fig. 4). Probability distributions were fitted to observations of these variables using the method of L-moments, with observed rainfall events being limited by a minimum WSA (1 mm) and DSD (60 min). Synthetic rainfall time series were then generated by producing random variates of these distributions.

Additionally, the temporal distribution of rainfall within a wet spell is described by a double exponential function conditioned on the wet spell time to peak (WSTP - modelled using a uniform distribution), wet spell peak intensity (WSPI - modelled using a copula, see below), and WSA (Fig. 5). Full details of the model can be found in Callau Poduje and Haberlandt (2017), with the following alterations which have been found to provide a better model performance, especially for regionalisation:

- a two parameter Khoudraji Gumbel copula describes the dependence between WSA and WSD,
- a two parameter Tawn copula describes the dependence between WSD and the ratio WSPI:WSA,
- the three parameter Weibull distribution is used instead of the four parameter Kappa distribution for the variables DSD and WSA, being more robust in a regionalisation setting.

Both copulas were fitted using the maximum pseudo-likelihood method. In total, the model consists of 19 parameters, fitted separately for summer (April-September) and winter (October-March).

### 3.3.2 Method Disagg: cascade model

Another possibility to generate high-resolution rainfall time series is to disaggregate daily time series, which generally exist for longer time periods and with higher station densities. For this study the micro-canonical cascade model after Müller-Thomy (2019, 2020) was applied due to its performance in previous studies (e.g. Müller and Haberlandt, 2015, 2018). The general cascade model scheme of disaggregating one coarse time step into "b" finer time steps is illustrated in Fig. 6. From daily rainfall amounts three time steps with 8 h duration are generated (b=3). For all further disaggregation steps b=2 is applied,

resulting in temporal resolutions of $\Delta t=\{4h, 2h, 1h, 30 \text{ min}, 15 \text{ min}, 7.5 \text{ min}\}$. To achieve $\Delta t=5$ min, a linear transformation is applied. More precisely, rainfall amounts of the $\Delta t=7.5$ min level are distributed uniformly on 2.5 min time steps, which are subsequently aggregated to 5 min time steps. The rainfall amount is conserved exactly in each disaggregation step. For a detailed description of the cascade model the reader is referred to the description of the preceding cascade model in Müller-Thomy (2020).

The cascade model parameters are estimated data-driven by the aggregation of observed 5 min time series of the recording stations available in each density scenario (no parameter calibration or optimisation was carried out). For the daily time series serving as starting point for the disaggregation, the aggregated 5 min time series of all 159 stations are used, independent of the applied density scenario. To investigate how suitable the disaggregation of daily information is for unobserved locations, a regionalisation of the daily precipitation is done prior to the application of the cascade model. Here, an ordinary kriging

approach (OK) was used for the regionalisation of daily rainfall time series from 2007-2016. The interpolation uses an isotropic exponential variogram as in Eqn. 6, a minimum of 4 neighbours and a maximum of 12 neighbours within a radius of 150 km.

$$\gamma(h) \;=\; c_0 \;+\; c\left[1 - exp(-\frac{h}{a})\right] \tag{6}$$

where $c_0$ is 0.4, $c$ is 1.5 and *aeff* is 186280 m.

### 3.4 Regionalisation schemes and station selection

In order to adequately test the performance of the different methods, all regionalisations were performed in a cross-validation mode at varying station densities. Five station density scenarios were chosen – 20% (N=32), 40% (N=64), 60% (N=96), 80% (N=128) and 100% (N=159) of the total available station count. Stations were selected at random and are consistent across the different testing methods presented above. Important to note is that the cross validation is only performed on the stations chosen at the 20% level (N=32) – the higher densities only add further supplementary stations available for the regionalisation,

with the assumption that the regionalisation performance will improve at a higher available station count. Furthermore, 20 realisations were performed for each density scenario to minimise the influence of station selection on the results. This scheme is illustrated graphically in Fig. 7.

## 3.5 Effect of time series length on the calculation or mean annual R

One further research question of interest is how long a rainfall time series must be in order to achieve a stable (i.e. robust) result of mean annual rainfall erosivity. The importance of such a research question lies in the fact that rainfall series used to calculate annual erosivity are often of limited length. Too short time series could lead to uncertain estimations of rainfall erosivity, which could affect the estimated soil erosion (i.e. under or overestimation).

Using the alternating renewal rainfall model described in section 3.3.1, we investigated how many years of data are needed in order to obtain stable estimations of the annual rainfall erosivity. For this purpose, 18 stations with the longest observation time series length (mean = 22 years) were selected for a more robust model fitting. For each station, the ARM model described in section 3.3.1 was fitted. Afterwards, 200 years of synthetic rainfall time series were generated 50 times for all 18 stations. The effect of time series length was investigated by calculating mean annual rainfall erosivity for each station and for each realisation using separately 2, 5, 10, 20, 30, 40, 50, 60, 70, 80, 90, 100, 120, 140, 160, 180 and 200 years of data. Results within ±20% of the mean annual erosivity calculated for 200 years were considered stable. The 20% threshold value was selected after checking the variability in the mean annual rainfall erosivity for specific stations and after comparing differences in mean annual rainfall erosivity values among stations. Hence, for most of the stations the maximum differences in the annual rainfall erosivity for specific years exceeded 20% while approximately ½ of the years the annual rainfall erosivity values were within the 20% range.

## 3.6 Evaluation criteria

To assess the relative performance of the four different methods (Fig. 3), three different evaluation criteria were chosen. The first is the Pearson correlation coefficient $r_{xy}$ between the annual rainfall erosivity calculated from observed rainfall and the estimated mean annual erosivity, i.e. regionalised by the different methods, at a given station density scenario across all realisations and the 32 reference stations. It is worth remembering that this measure is an indication of linear correlation only. The second is the relative root mean square error (RMSE) of the mean annual rainfall erosivity for a given station density scenario across all realisations and stations, given by equation 7:

$$RMSE = \sqrt{\frac{1}{N}\sum_{n=1}^{N}\left[\frac{y-x}{x}\right]^2} \qquad (7)$$

where $x$ refers to mean annual rainfall erosivity estimated from observations, $y$ is the estimated mean annual rainfall erosivity estimated by the different methods proposed in this work and $N$ is the number of stations. The third is the median relative bias $rB_{med}$ of the mean annual erosivity. For a given station density scenario the relative bias $rB$ is calculated over all 32 stations for one realisation, and the median relative Bias $rB_{med}$ is provided as summarizing result (referred to as relative Bias in the following):

$$rBmed = median(\frac{y-x}{x}) \qquad (8)$$

## 4 Results and discussion

### 4.1 Mean annual erosivity R

The main focus of this study was to evaluate the performance of the four different tested methods in reproducing observed mean annual erosivity as input to the USLE-type models. It is the most relevant attribute from the perspective of the soil erosion modelling community. For the direct regionalisation of erosivity (Direct-R), EDK with mean annual precipitation ($P_{mean}$) as external drift was found to provide the best performance. Direct regionalisation provides best results overall, both in terms of accuracy and precision. The Pearson correlation (Fig. 8) by this method is larger than for the other three methods,

which show more or less the same result. Its root mean square error is also lower than the other methods (Fig. 8). The $rB_{med}$ is the only performance criteria where this method is not clearly better (Fig. 8). In terms of $rB_{med}$ the Direct-R method is followed by Direct-P. ARM and Disagg in most cases yield relative bias larger than 10 and 20%, respectively. In terms of relative RMSE and Pearson Correlation, the ARM and Disagg methods performed relatively similarly. The median result over all realisations showed slightly better results for the direct regionalisation of rainfall (Direct-P) compared to stochastic rainfall models,

although the range of results is greater than for Direct-R. The disaggregation model performed worse than the other methods with a >20% underestimation of mean annual erosivity. However, the Disagg results were found to be sensitive to the applied 'measuring resolution' of the generated time series (please see the supplement for details; Fig. S1), which could be represented better by simple modifications (Müller-Thomy, 2020).

With increasing station density, the median result was generally improved, in particular for the Pearson correlation (Fig. 8).

This is less noticeable for $rB_{med}$, as bias likely indicates the inherent ability of the underlying method to replicate the mean annual R (Fig. 8). Moreover, it seems that for ARM and Direct-P the results change greatly between realisations (i.e. larger spread of the results), meaning that the station setting is more decisive, with Direct-P being the most affected. The Direct-R and Disagg methods seem to be more robust in terms of station setting, which indicates their usage for regions with low station densities. Thus, it seems that for areas similar to that of Lower Saxony (i.e. relatively flat areas with small changes in erosivity),

the Direct-R method is preferred in case that one needs to estimate the rainfall erosivity for the ungauged locations. The application of this method is also the simplest of the four methods (Fig. 3), especially when compared to the ARM and Disagg methods, since these two methods require development of the stochastic rainfall models and estimation of the model parameters. Thus, in terms of mean annual rainfall erosivity, the use of stochastic rainfall models did not provide an improvement of the results in terms of performance in space (Fig. 8).

In terms of the number of stations needed to provide good estimates of rainfall erosivity for the ungauged locations, it is clear that higher station density yields better results, which was expected (Fig. 8). Thus, Direct-R showed decreasing performance with decreasing station density, D100>D80>D60>D40>D20 (Fig. 8), with somewhat diminishing improvements in performance above D80. Similar conclusions can be made for the Direct-P method while for the ARM and Disagg this relationship was not so evident for the $rB_{med}$ and RMSE (Fig. 8). Therefore, even for topographically relatively homogeneous

areas (compared to for example Alpine area) such as Lower Saxony, the accuracy of rainfall erosivity maps greatly depend on

the number of stations used for the interpolation. Hence, it is clear that estimation of rainfall erosivity for ungauged areas is in all cases exposed to some degree of bias due to the interpolation (Fig. 8). Even in cases where the density of stations is high (e.g., D80 or D100 scenarios), there was a slight bias (few percentage) in estimation of the rainfall erosivity for ungauged locations using the Direct-R and Direct-P methods while for ARM and rainfall disaggregation model the median relative bias is larger than 10% and 20%, respectively (Fig. 8). Additionally, it should be assumed that the estimation bias for ungauged locations within topographically complex terrain would be even higher. Therefore, these results indicate that the density of gauging stations should be as high as possible in order to obtain optimal rainfall erosivity estimates.

## 4.2 Erosive event characteristics

Although the main focus of the study was the calculation of the mean annual erosivity R, in some specific applications more details about the rainfall erosivity at a location might be required and characteristics of erosive events are often studied and compared (e.g., Bezak et al., 2021b). This information is not directly used as input to the USLE-type models, however, studies that investigate changes in erosivity patterns in time (i.e. climate change studies) might be interested in changes in the number of erosive events or erosive events characteristics. In such a case, the Direct-R method cannot be used since this method is only able to provide estimates of the mean annual rainfall erosivity and does not provide a time series of erosive events. Naturally, a similar procedure as in the case for Direct-R (section 3.2.1) could be used for any other variable of interest (e.g. number of erosive events, mean duration of the erosive events). However, the EDK setup used for mean annual R is unlikely to be optimal for other variables, which means that an optimal EDK setup would need to be investigated for each variable of interest separately and thus could be considered relatively time consuming.

However, three of the four methods presented in this study, namely Direct-P, ARM and Disagg, are able to reproduce erosive events themselves from which the mean annual R is calculated. How well the three methods reproduce erosive event characteristics is explored in this section. Fig. 9 displays the median relative bias of different erosive event variables. The mean annual event count and precipitation sum is best represented by the Direct-P method. Whereas the median error is close to zero, the range of errors is well over 100% when one considers outliers. The Disagg method represented the event count well but underestimates the annual volume slightly, whereas the ARM method underestimated both the annual count and volume (Fig. 9). Performance improved slightly for all methods with increasing station density (not shown).

For both the mean event duration and volume (Fig. 9), the results again show that the Direct-P method was the most efficient. The Disagg method significantly overestimated the event duration and at the same time underestimated the event volume. However, in the case of using the Disagg method the station density showed less of an effect on performance compared to the other methods (not shown here). An overestimation of the erosive events duration was to be expected as being previously identified by Jebari et al. (2012) with overestimations higher than 40%. Here the overestimations are higher as the disaggregation inherits the errors caused by the regionalisation of the daily rainfall. Because of the unbiased estimator, OK tends to smooth the spatial structures of the rainfall, leading to overestimation of the low intensities (explaining the longer

event durations) and underestimation of the extreme ones (explaining the lower event volumes). The performance of the disaggregation may improve if another regionalisation method that better captures the temporal variability is employed.

The ARM method underestimated both mean event duration and mean event volume, which explains the underestimation in both annual number of events and their volume. Thus, it can be seen that Direct-P outperformed the ARM and Disagg methods not only in terms of annual rainfall erosivity but also in terms of specific events characteristics. In case one would like to obtain erosive event characteristics for ungauged sites, the Direct-P method should be preferred since it was able to produce acceptable results and yielded better performance compared with the two tested stochastic rainfall models.

Moreover, it should be noted that other available stochastic rainfall models could perhaps yield better performance in comparison to the selected stochastic rainfall models (see section 1 for examples). The scope of the study limited the selection of stochastic rainfall models to the two presented in this study only. Since any rainfall model has its own unique strengths and weaknesses, it is possible that some other non-tested model could yield better performance in terms of reproducing maximum 30-minute rainfall intensities, which are directly used for the estimation of the rainfall erosivity (section 3.1).

**4.3 Annual rainfall erosivity – data length sensitivity**

Despite the fact that Direct-R and Direct-P methods yielded better performance than the evaluated stochastic rainfall models, the benefit of the latter is the ability to generate long time series of arbitrary length of high-resolution data for unobserved locations. The goal of this section is to investigate how long the synthetic time series should be in order to obtain a stable estimate of the mean annual rainfall erosivity, which is most frequently used as an input to the soil erosion models (Panagos

et al., 2015; 2017). Thus, this information is relevant for the soil erosion modellers in order to evaluate the impact of potential bias in the assessment of the mean annual rainfall erosivity on the soil erosion modelling results. Using the ARM model and the methodology described in section 3.5 it was investigated how many years of data is needed in order to obtain stable annual rainfall erosivity estimation. Important to remember is that the ARM model performance for this task is better than what was shown in sections 4.1 and 4.2 as here the ARM model was fitted directly to observations and not regionalised. Fig. 10 shows

results of this investigation. It can be seen that the variability between different realisations was quite high (Fig. 10). Moreover, investigation of the intersection between the 5 & 95% realisation quantiles (i.e. with the aim to exclude potential extremes) and the ±20% interval of the mean annual rainfall erosivity (calculated from the full 200-year time series) indicates that in the case of the 5% quantile (Fig. 10), in most cases 60 years of data were needed in order to obtain a value within this ±20% interval (Fig. 10). This indicates that calculations of the annual rainfall erosivity for soil erosion modelling using USLE or

RUSLE (e.g., Renard et al., 1997; Panagos et al., 2017) using only a limited sample size (e.g., less than 5 or 10 years) will likely result in a greater than ±20% difference to the long-term mean. Consequently, a similar impact (i.e. over or underestimation) on the calculated soil erosion rates will be obtained if one applies the RUSLE equation for the prediction. Moreover, it should be noted that the results are to some extent sensitive to the selection of the threshold (i.e. 20% value). More specifically, using lower threshold values would result in needing longer time series to obtain stable annual rainfall

erosivity estimates. Conversely using higher threshold values would require shorter time series.

## 5 Conclusions

This study evaluated four methods that can be used to estimate the annual mean rainfall erosivity (R) in space. Based on the presented results the following conclusions can be drawn:

1. For the mean annual rainfall erosivity both tested direct regionalisation methods (Direct-R and Direct-P) outperformed (Fig. 8) the tested stochastic rainfall models (ARM and Disagg), with slightly better results for Direct-R. Furthermore, in terms of method complexity, Direct-R can be regarded as the simplest since it does not require the fitting of any model parameters. Differences among tested methods were relatively large, for example, in relative bias up to 25%.

2. The main drawback of the Direct-R method is that it cannot be used to estimate the number of erosive events or mean event duration without applying the model to every variable separately (e.g., number of erosive events, annual rainfall erosivity). This information is sometimes additionally required in erosivity studies, although it is not directly used by the USLE-type models. Therefore, the Direct-P method has the advantage that it is able to generate high-resolution time series of erosive events for ungauged sites. Therefore, information about the number of erosive events and the characteristics of erosive events can be determined as well. In terms of the characteristics of the erosive events, the Direct-P method yielded better performance than both tested stochastic rainfall models.

3. Both rainfall generators have proven their applicability in the field of soil erosion modelling since they are able to produce long synthetic series of the high-resolution data, which can be used to calculate stable rainfall erosivity estimates.

4. The cross-validation methodology using multiple density scenarios (Fig. 7) indicated that all methods performed slightly better with increasing station density. However, interpolation of rainfall erosivity for ungauged locations will in case of the Direct-R and Direct-P methods introduce some bias (~5%), also in the case of having very high station density (Fig. 8). In case of ARM and Disagg methods, this bias can be larger than 10% (Fig. 8). Even more significant differences could be expected in case of topographically more complex areas. Hence, station density should be as high as possible to obtain optimal rainfall erosivity estimates for ungauged locations.

5. Investigation of the impact of time series length on the annual rainfall erosivity for 18 stations was additionally carried out using the ARM model. More than 60 years of data was required in the case that one would like to obtain rainfall erosivity estimates within 20% of the actual long-term mean annual rainfall erosivity.

Thus, this conclusion is of critical importance for soil erosion studies where rainfall erosivity estimates are used as input, since in most cases the high-resolution data used to estimate rainfall erosivity is much shorter than 60 years. So, in cases where only 5-10 years of observed rainfall data is available, the estimated mean annual rainfall erosivity can be up to ±100% in comparison to the actual long-term mean annual rainfall erosivity, which can be reduced by the application of one of the here used rainfall generators.

It should be noted that the approaches presented in this paper should be applied and tested for further case studies with different rainfall and topographical characteristics than for Lower Saxony, which is mostly flat and without major orographic obstacles.

Additionally, some study limitations and lessons learned can also be made based on the presented results and conclusions such as that resolution of the measurement device, which has evolved in the recent decades, has a significant effect on the calculated rainfall erosivity and relative bias (Supplement; Fig. S1).

**Funding**

The results of the study are part of the bilateral research project between Slovenia and Germany "Stochastic rainfall models
for rainfall erosivity evaluation" and research Programme P2–0180: "Water Science and Technology, and Geotechnical Engineering: Tools and Methods for Process Analyses and Simulations, and Development of Technologies" (P2-0180) that is financed by the Slovenian Research Agency (ARRS). Hannes Müller-Thomy has been financially supported by the DFG e.V., Bonn, Germany, as a Research Fellowship (MU 4257/1-1). Additionally, part of the results were also obtained in the scope of the bilateral project between Slovenia and Germany "Validation of precipitation reanalysis products for rainfall-runoff
modelling in Slovenia (PRE-PROMISE)", funded by the German Federal Ministry of Education and Research (BMBF).

**Acknowledgment**

We would like to acknowledge German Weather Service (DWD) for provide data used in this study. Also, Nadav Peleg, Mark Silburn and one anonymous reviewer are acknowledged as well as Greg Hancock as Associate Editor.

**Competing Interest**

Authors declare no conflict of interest.

**Author contribution**

All authors developed the concepts of the manuscript. R.P. conducted most of the calculations with support of the B.S., H.M.T., U.H. and N.B., N.B. drafted the first version of the manuscript, all authors contributed to writing and editing of the manuscript.

**Code and data availability**

Data can be requested from the German Weather Service (DWD), the code used in this study are freely available upon request from the first author.

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

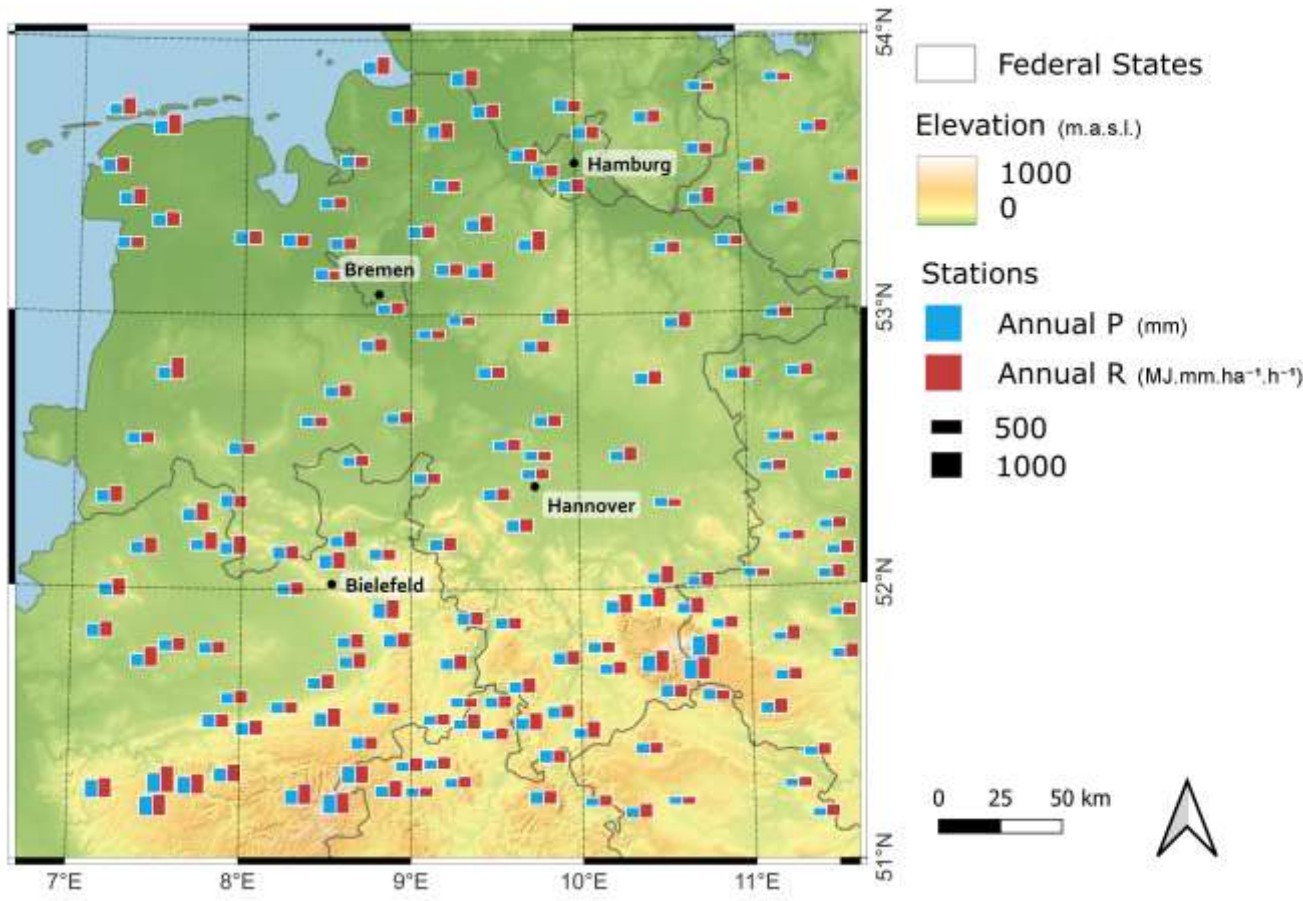

**Figure 1: Location of all recording stations (N=159). The bar plots indicate the observed annual precipitation volume (blue) and erosivity factor (red) .**

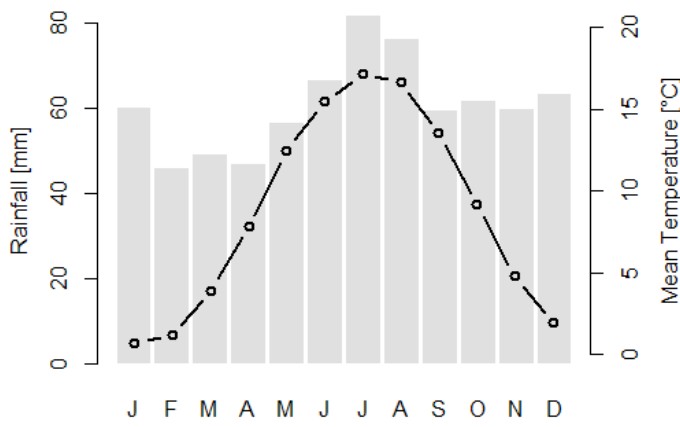


**Figure 2: Long-term climate (1881-2019) averaged across the German federal state of Lower Saxony that is investigated in the scope of this study. Source: German Weather Service (DWD) Climate Data Center (CDC) (CDC, 2020).**

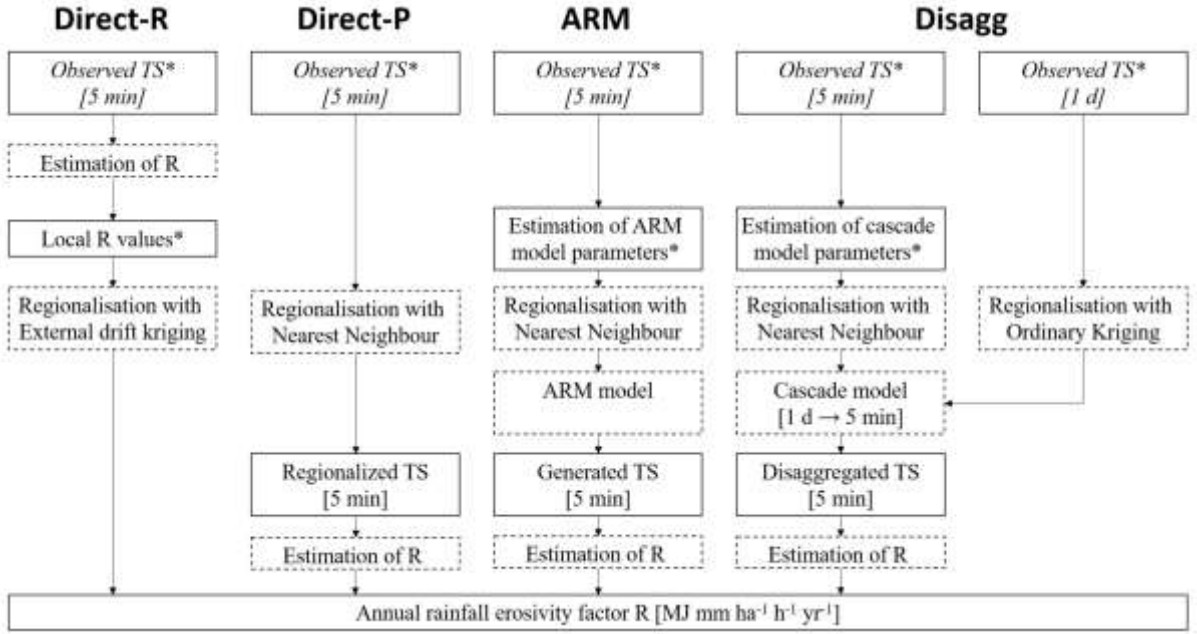

**Figure 3: Overview of the estimation of the annual rainfall erosivity (R) including all applied methods (dashed boxes) and data sets (solid boxes, italic written represent input data sets) ['TS' are rainfall time series; '*' indicates data sets affected by the station density scenarios].**

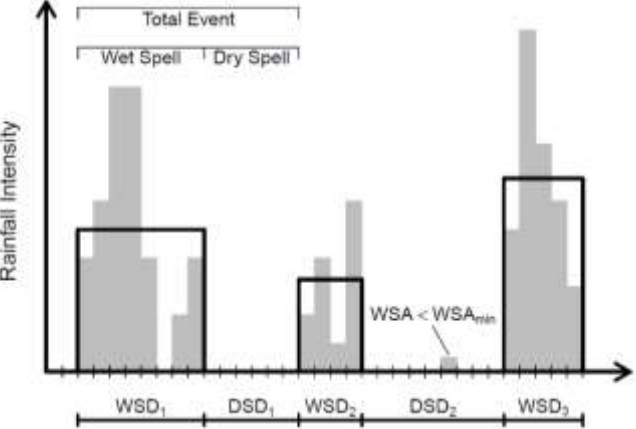

**Figure 4: Schematic of the external structure of the ARM model. The black boxes describe rainfall events derived from observations.**

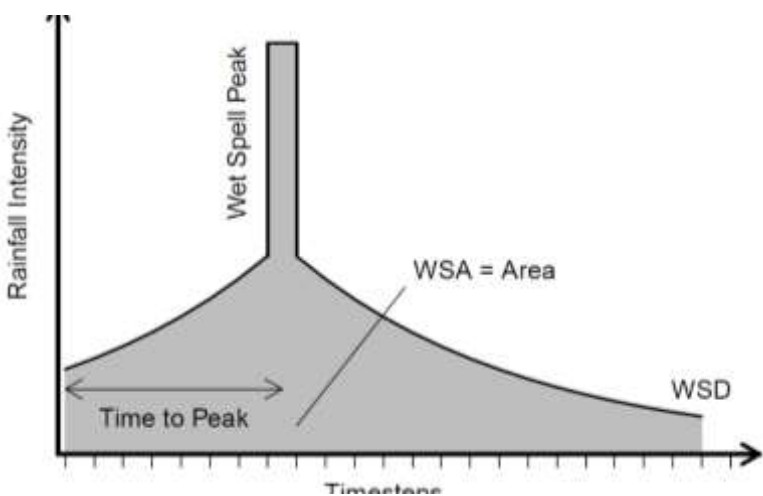

**Figure 5: Internal structure of ARM model according to the Callau Poduje and Haberlandt (2017).**

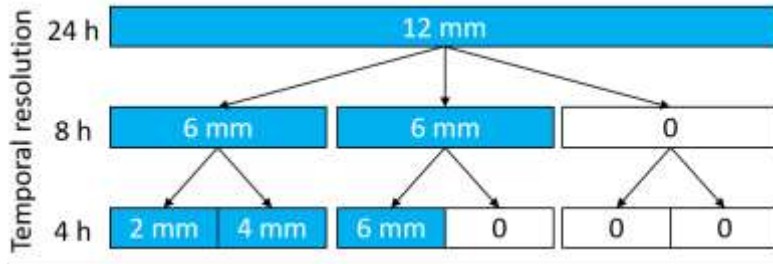


**Figure 6: General scheme of the cascade model for the first two disaggregation steps with exemplary rainfall amounts for a daily total of 12 mm (blue boxes = wet time steps).**

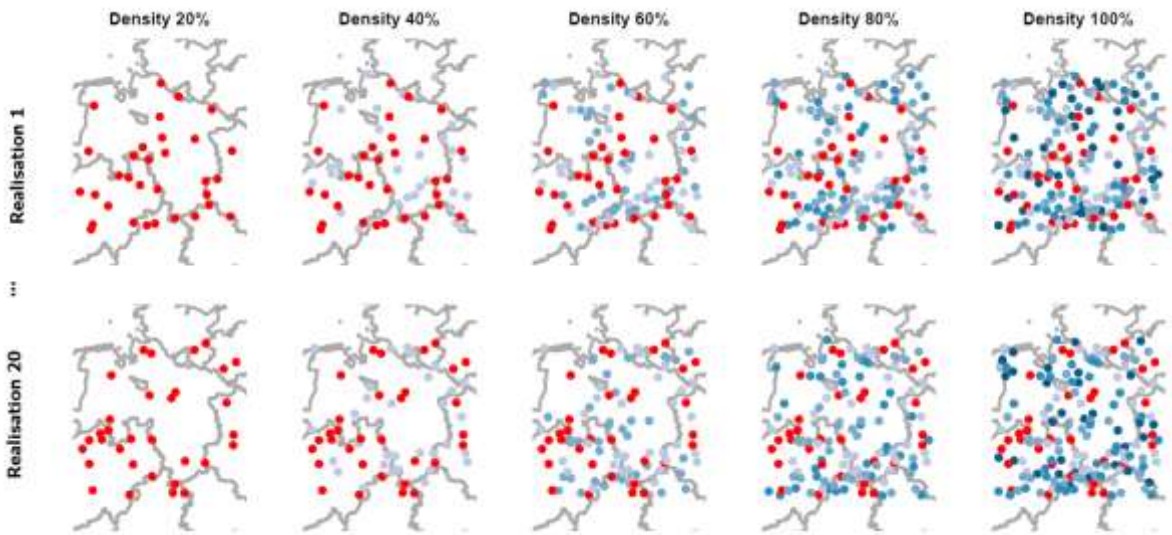

**Figure 7: Five different station density scenarios shown for two realisations. The red stations are used for the cross validation for all station densities. The additional stations (blue) provide supplementary information for the regionalisation, with darker shades added to previous scenarios.**

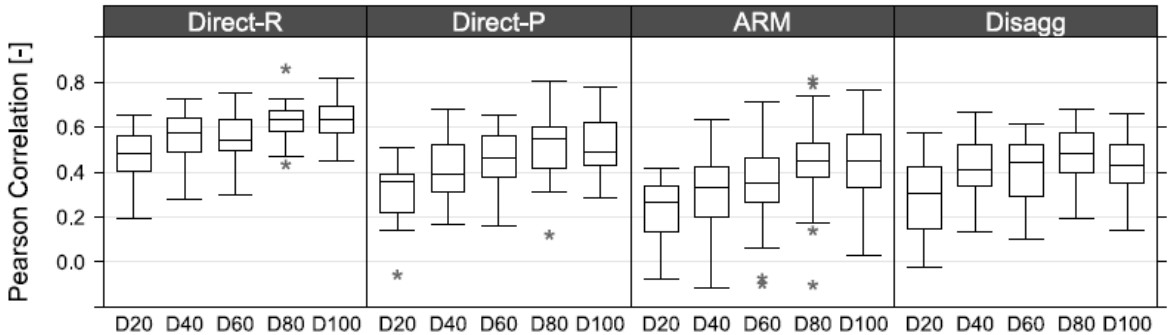

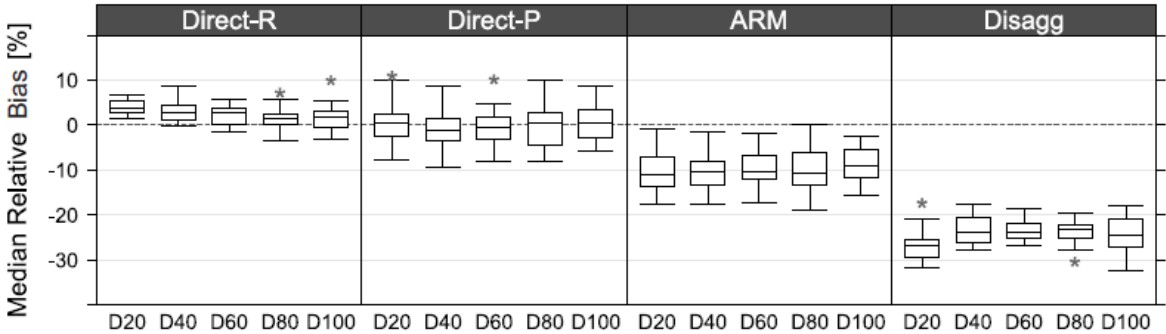

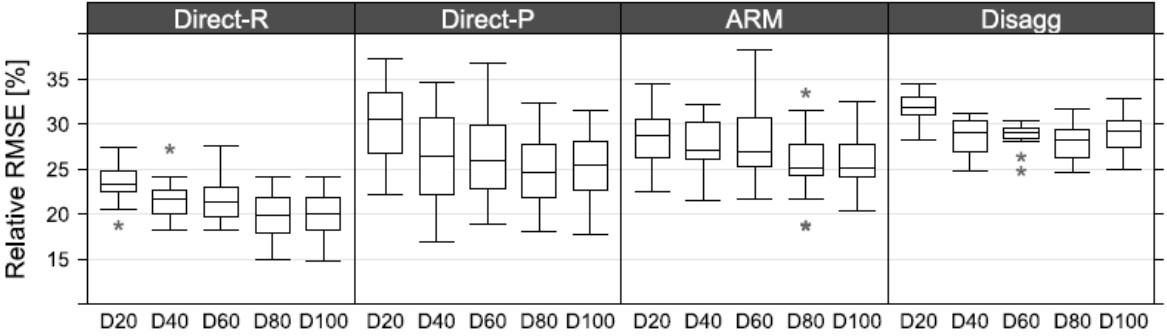

**Figure 8: Pearson's Correlation Coefficient, Relative Bias and Relative RMSE results for the four tested methods. Box plots show the relevant statistic for the 20 realisations (N = 20) at each density scenario (e.g. D20 corresponds to the 20% station density scenario).**

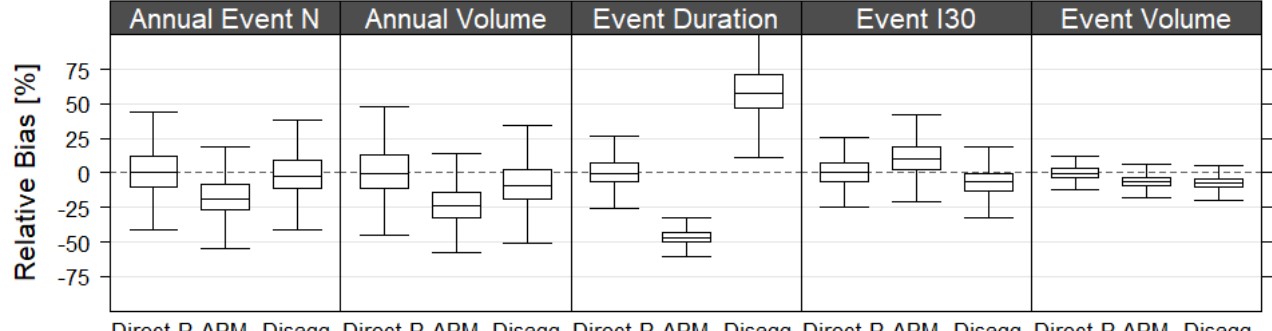

**Figure 9: Relative bias of all stations (N=32) for all realisations (N=20) and density scenarios (N=5) for different erosive event variables (for each box plot, N = 3200). Annual number of erosive events, annual number of all erosive events volume, erosive event duration, maximum event I30 rainfall intensity, and erosive event volume are shown. Outliers are excluded for clarity.**

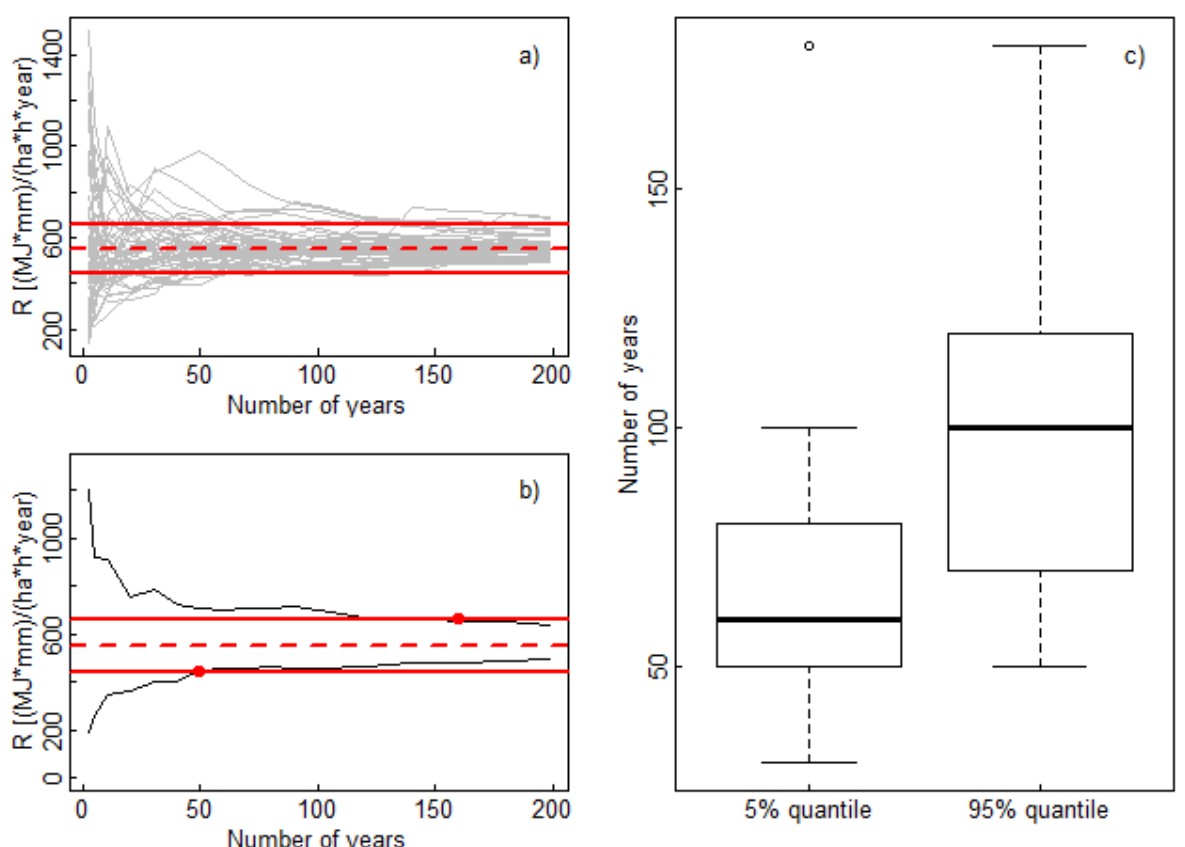

**Figure 10: Relationship between mean annual rainfall erosivity and number of years used in the rainfall erosivity calculation. a) shows 50 different realisations (grey lines) together with the mean annual value calculated using all 50 realisations with 200 years of data (dotted red line) and ±20% of this mean value (solid red line). b) shows 5% and 95% quantile values (black line) calculated based on 50 realisations shown in a) and intersections between these quantile values and ±20% interval. c) shows boxplots of intersection points for 17 stations for the 5% and 95% quantile. a) and b) are shown for station ID = 10113 (Nordeney).**

**Table 1: Median values over all reference stations (N=32) and realisations (N=20). Simulated values are given for the 100% density scenario.**

|  | Annual Erosive Events [-] | Annual Erosivity R [MJ.mm.ha$^{-1}$.h$^{-1}$] | Mean Event I30 [mm.h$^{-1}$] | Annual Erosive Rainfall Volume [mm] | Mean Event Duration [hrs] | Mean Event Volume [mm] |
|---|---|---|---|---|---|---|
| **Observed** | 14.5 | 642.7 | 10.86 | 303.2 | 22.5 | 21.1 |
| **Direct-R** | - | 653.8 | - | - | - | - |
| **Direct-P** | 14.6 | 647.7 | 10.89 | 303.6 | 22.6 | 21.0 |
| **ARM** | 11.6 | 580.4 | 11.92 | 227.1 | 12.0 | 16.8 |
| **Disagg** | 14.2 | 475.5 | 10.09 | 277.8 | 35.9 | 19.6 |