# Peer review of "Comparison of rainfall generators with regionalisation for the estimation of rainfall erosivity at ungauged sites"

_Earth Surface Dynamics, 2022_

## Referee Comment (RC1)

**General**

The study by Pidoto et al. provided valuable insight into the potential use of rainfall generators and disaggregation models to estimate rainfall erosivity. The paper is clearly written and well structured. As I mentioned in the specific comments below, I do, however, miss a brief discussion about the implications of the results. Methods are scientifically sound, and results are presented in appropriate figures and tables. It would be good to see the results of the stochastic models (rainfall properties) before the results of the rainfall erosivity and describe how the results of the first affect the results of the second in the discussion section. The majority of my comments below require text edits (no further model runs required), and I think the authors will have no problem revising the paper accordingly, should they decide to do so.

Sincerely,
Nadav Peleg

**Specific comments**

Line 46. You might want to add another reference from our recent work (https://doi.org/10.1016/j.geomorph.2021.107863). We also tailor there between the spatial structure of rainfall and rainfall erosivity skill (see Fig. 9, for example), which you may find interesting.

Line 55. This introduction emphasizes the importance of high temporal resolution data/simulations (e.g. 5 min). But what about spatial resolution? It might be worth adding a sentence about this aspect as well (e.g. do you need gridded, 2-D, stochastic models, or 1-D is sufficient?). It might be worth adding a sentence about this aspect as well (e.g. do you need gridded, 2-D, stochastic models, or 1-D is sufficient?).

Lines 99-100. There is no definition for $M$.

Equation 3. Replace the commas with points.

Equation 7. Since Pearson's correlation is well-known, this equation does not need to be presented.

Results section. You begin by presenting the results of the mean annual erosivity. The erosivity, however, is calculated using rainfall intensity at intervals of 30 minutes (i.e. equation 1). Perhaps it would be better to start by presenting the results of the simulation (the disaggregation and the ARM) in reproducing the 30-minute maximum rainfall? For example, I wonder if the same bias in erosivity (10% and 20%, line 226) will also be visible in the rainfall simulations.

Lines 270-272. Could you show this in a figure? Perhaps as supplementary material?

Lines 274-275. This comment is related to the one I made above. Before discussing erosivity, it would be useful to discuss the results of ARM and Disagg. Underestimate by how much? What about the intense short-duration rainfall (30 min) - are they also underestimated?

Discussion. There does not seem to be any discussion about the fact that only two stochastic rainfall models (a rainfall generator model and a rainfall disaggregation model) were tested. I am not suggesting the authors run additional models, but to point out that other stochastic models/methods can result in a different rainfall time series, which, for example, can better simulate the I30 and thus better reproduce rainfall erosivity. Moreover, I suggest the authors add a paragraph discussing the potential implications of these methods in ungauged areas. Despite the fact that it is mentioned at the beginning of the manuscript, I am missing some discussion points and conclusions.

Figure 1. Could the resolution be improved? How important is the information on the summer/winter precipitation? It would be nice to see the RUSLE information on the map as well (maybe by using different colors for each point?) so the readers can see how rainfall erosivity is distributed within the domain.

Figure 3-6. Here too, the figure resolution needs to be improved.

Figure 5-6. I have seen these figures several times... Maybe they can be presented as supplementary material instead of being included in the text?

Line 419. The reference's year is incorrect - it should be 2017.

---

## Referee Comment (RC2)

[referee-annotated manuscript omitted]

---

## Referee Comment (RC3)

The manuscript used 5-min and daily data from northern Germany to evaluate the accuracy of 4 different methods to estimate the rainfall erosivity (R) for sites where high temporal resolution rainfall data are not available. The manuscript is structured well, and mostly readable. The authors should be commended for being thorough and rigorous in their data analysis.

Presentation is ok, English expressions need attention to improve the quality of the manuscript. I have edited the abstract for the authors. Similar effort is probably needed throughout the manuscript, especially with respect to the tense. When we describe what we did, we need to use the past tense. Use the present when we cite other people's work, especially their observations and conclusions, because they have been published and, in a sense, made permanent.

Specific comments

Fig. 1 – Poor quality. No longitude/latitude, no scale. So not all stations are located in Lower Saxony.

Fig. 3 – Use 'Computed R' for the first dashed box top left?

Fig. 4 – Y-axis. If there is no scale, what is the point of having the unit?

Ditto Fig. 5

Line 38 - ha$^{-1}$ h$^{-1}$

Line 57 – What do you mean by 'high frequency'?

Around Line 60

To use the USLE/RUSLE, the following is needed: the R value, and monthly or half-monthly distribution of erosivity, and 10-year event EI30 values. While storm event characteristics are relevant, they are not needed, strictly speaking. Re-define and specify the research objectives. If event EI30 is included, some of the 4 methods are automatically disqualified. In other words, it is necessary to clearly define what aspect(s) of the rainfall erosivity to be estimated and the methods that can be used for that estimate.

Around Line 75

Fig. 1 indicates that stations outside of Lower Saxony were also included in the study. It is better to define the study area using latitude and longitude boundary to include Lower Saxony within that boundary.

10-year could normally be short for computing the R factor values. What is the interannual variability of annual precipitation and annual EI30 values among the recording stations with 5-min data?

Line 107

I understand that ',' is used for decimal places in many European countries. Not sure about the journal policy on this. I'd like to see the authors use 0.29, 0.72 etc as they appear in the original reference cited by the authors for an international readership.

Around Line 125

The authors include many covariables for EDK. May I suggest author include the mean annual precipitation (P) as one of the additional variables in Equation (5) because P is widely and reliably available and we the P and R are well related around the world listed below.

Renard, K. G., & Freimund, J. R. (1994). Using monthly precipitation data to estimate the *R*-factor in the revised USLE. *J. Hydrol., 157*(1-4), 287-306.
Yu, B., & Rosewell, C. J. (1996). A robust estimator of the *R*-factor for the Universal Soil Loss Equation. *Trans. ASAE, 39*(2), 559-561.
Zhu, Z., & Yu, B., (2015). Validation of rainfall erosivity estimators for mainland China, *Trans. ASABE*, 58(1), 61-71.

Around Line 155

How many parameters to be calibrated using observations for ARM? Do the parameters vary monthly or seasonally? How were parameter values estimated?

Line 18 – Linear transformation. How was this achieved? Did you have to extrapolate from 15-min and 7.5-min down to 5-min?

Again, how many parameters involved in the disaggregation method, how were they estimated, i.e. method of estimation?

Around 175

EDK was used to spatially interpolate the R factor. Why was this not used with covariables to interpolate daily rainfall amount before disaggregation?

Reference:

Line 194
'18 stations with the longest' record length: How long were they in number of years? Compute annual EI30 values to shed light on its underlying interannual variability to provide some empirical support for selecting 20% as a criterion to define a 'stable' estimate.

Result section

The result in relation to R is fine, but there is a need to justify the way erosive events were selected and aspects of these events were defined. Again, if the intent of the manuscript is to prepare the best possible input for the USLE/RUSLE, one needs to use events as defined in the USLE/RUSLE (Renard et al. 1997). If one selects events mostly for the sake of testing and comparing different rainfall interpolation approaches, it is useful to spell this out as one of the distinct research objectives.

Conclusions

No. 1 Again, one questions whether we need to have the number of erosive events and event duration for the USLE/RUSLE. This depends on the research objectives. If we focus on event-level EI30 values, and other aspects, it is not even fair to include the Direct-R in the comparison.

No. 5 This needs to be assessed in the context of the underlying interannual variability of EI30 values for the region.

Abstract with track change as an example:

Rainfall erosivity values are required for The assessment of rainfall erosivity is one of the main inputs in determining soil erosion prediction. To calculate the mean annual rainfall erosivity (R), long-term high-resolution observed rainfall datatime series are required, which are often not available. To 15 overcome the issue of limited data availability in space, four methods wereare employed and evaluated: the direct regionalisation of R, the regionalisation of 5- minute rainfall, the disaggregation of daily rainfall into 5- minute timesteps, and the use of a regionalised stochastic rainfall model. In addition, the minimum recordtime series length necessary to adequately estimate R wasis investigated for. The impact of station density is considered for each of the 4 methods. The study wasis carried out using 159 recording and 150 nonrecording (daily) rainfall stations in the federal state of Lower Saxony, Germany. Results show that the direct rregionalisation 20 of the mean annual erosivity is leads to the best results in terms of relative bias and relative root mean square error (RMSE). This is
followed by the regionalisation of the 5- minute rainfall data, which yields better results than the rainfall generation models,
 namely an alternating renewal model (ARM) and a multiplicative cascade model (Disagg). However, a key advantage of using

regionalised rainfall models is the  generat erosive event

characteristics, which is not possible with direct regionalisation of R. Using the stochastic ARM

 more than 60 years of data is needed in most cases  to reach a stable estimate of the annual rainfall erosivity. Estimation

of soil erosion based on only 5 or 10 years of data can lead to uncertain R values. Such short time series are often used when

regionalisation is applied. Moreover, it was also found that temporal resolution of measuring device has a significant effect on the

rainfall erosivity and coarser data resolution can lead to high relative bias.

---

## Author Comment (AC1)

Brunswick, 22nd of June 2022

**TO:**
**Editorial Office**
**Earth Surface Dynamics**

Dear Editorial Office,

Please find enclosed the revised version of the original manuscript entitled "*Comparison of rainfall generators with regionalisation for the estimation of rainfall erosivity at ungauged sites*" authored by R. Pidoto et al.

We would like to thank the editor and the reviewers for their time and efforts in reviewing our manuscript, for the constructive comments and observations that can help us to bring the manuscript up to ESD standard, and for the opportunity to address the highlighted concerns and suggestions in this resubmission.

In response to the reviewers' comments, we have conducted a major revision and incorporated new material in the revised version. Most notably: (1) we have additional discussion about multiple aspects such as performance for the ungauged locations, (2) we modified the aims of the study in order to better reflect the content, consequently conclusions were also modified  (3) we have modified several figures and introduced new material as proposed by the reviewers, and (4) several smaller corrections that were suggested by three reviewers were incorporated into the manuscript.

Please find below a detailed response to all of the reviewer's comments. All changes will be indicated with red coloured text in the manuscript.

Sincere regards on behalf of all the authors,

Hannes Müller-Thomy (as corresponding author)

**Reviewer #1 (Dr. Nadav Peleg)**

*R1C0 Comment*: The study by Pidoto et al. provided valuable insight into the potential use of rainfall generators and disaggregation models to estimate rainfall erosivity. The paper is clearly written and well structured. As I mentioned in the specific comments below, I do, however, miss a brief discussion about the implications of the results. Methods are scientifically sound, and results are presented in appropriate figures and tables. It would be good to see the results of the stochastic models (rainfall properties) before the results of the rainfall erosivity and describe how the results of the first affect the results of the second in the discussion section. The majority of my comments below require text edits (no further model runs required), and I think the authors will have no problem revising the paper accordingly, should they decide to do so

*R1C0 Response*: We would like to thank Dr. Nadav Peleg for reviewing our manuscript. We very much appreciated the encouraging comments and overall positive evaluation on our study.

As suggested, the discussion section was enhanced with focus to provide more details about importance of the study. Additionally, results of the stochastic rainfall models were added and a comparison with gauge-based data was done.

Point-by-point detailed responses to the specific comments are provided below. Thanks.

**Specific comments**

*R1C1 Comment*: Line 46. You might want to add another reference from our recent work (https://doi.org/10.1016/j.geomorph.2021.107863). We also tailor there between the spatial structure of rainfall and rainfall erosivity skill (see Fig. 9, for example), which you may find interesting.

*R1C1 Response*: The suggested reference was added.

*R1C2 Comment*: Line 55. This introduction emphasizes the importance of high temporal resolution data/simulations (e.g. 5 min). But what about spatial resolution? It might be worth adding a sentence about this aspect as well (e.g. do you need gridded, 2-D, stochastic models, or 1-D is sufficient?). It might be worth adding a sentence about this aspect as well (e.g. do you need gridded, 2-D, stochastic models, or 1-D is sufficient?).

*R1C2 Response*: Thanks for your remark. Additional explanation was added to the manuscript.

*R1C3 Comment*: Lines 99-100. There is no definition for M.

*R1C3 Response*: Noted with thanks. M is now defined among other variables.

*R1C4 Comment*: Equation 3. Replace the commas with points.

*R1C4 Response*: Thanks for your remark. We have corrected this typo. Thanks.

*R1C5 Comment*: Equation 7. Since Pearson's correlation is well-known, this equation does not need to be presented.

*R1C5 Response*: Thanks for the comment. We decided to remove equation 7 as proposed by the Dr. Nadav Peleg.

*R1C6 Comment*: Results section. You begin by presenting the results of the mean annual erosivity. The erosivity, however, is calculated using rainfall intensity at intervals of 30 minutes (i.e. equation 1). Perhaps it would be better to start by presenting the results of the simulation (the disaggregation and the ARM) in reproducing the 30-minute maximum rainfall? For example, I wonder if the same bias in erosivity (10% and 20%, line 226) will also be visible in the rainfall simulations.

*R1C6 Response*: Noted with thanks. Please note that the main idea of the paper was to compare four selected methods, hence we argue that first comparison of mean annual erosivity is the most meaningful while in the following sections we also discuss some of the detailed erosive event characteristics. Hence, as suggested we have also added some results about the 30-minute maximum rainfall as suggested by Dr. Nadav Peleg.

*R1C7 Comment*: Lines 270-272. Could you show this in a figure? Perhaps as supplementary material?

*R1C7 Response*: Thanks for your remark. At this stage no additional figure has been included.

*R1C8 Comment*: Lines 274-275. This comment is related to the one I made above. Before discussing erosivity, it would be useful to discuss the results of ARM and Disagg. Underestimate by how much? What about the intense short-duration rainfall (30 min) - are they also underestimated?.

*R1C8 Response*: Noted with thanks. We have extended Fig.9 by I30 for a visual interpretation. As mentioned before, the focus of the manuscript is on rainfall erosivity (which is interesting for the community), not on the single steps used for its determination. Although we understand the interest of the reviewer in these details, we think showing them would move the focus away from the approaches not based on rainfall generation. Hence, we stick to the visual comparisons we have and would like to avoid to provide detailed percentages for each bias/relative error.

*R1C9 Comment*: Discussion. There does not seem to be any discussion about the fact that only two stochastic rainfall models (a rainfall generator model and a rainfall disaggregation model) were tested. I am not suggesting the authors run additional models, but to point out that other stochastic models/methods can result in a different rainfall time series, which, for example, can better simulate the I30 and thus better reproduce rainfall erosivity. Moreover, I suggest the authors add a paragraph discussing the potential implications of these methods in ungauged areas. Despite the fact that it is mentioned at the beginning of the manuscript, I am missing some discussion points and conclusions.

*R1C9 Response*: Thanks for your remark. As suggested, additional discussion about the stochastic rainfall models was added (section 4.2). Moreover, some discussion about the interpolation of erosivity for ungauged locations has been added to the sections 4.1 and 5. The authors have previous experiences with the rainfall generators used in this study, but would feel uncomfortable to recommend other rainfall

generators for certain erosive event characteristics due to their missing experience with these models. References could be implemented as a compromise, but from a brief review the authors could not find some pointing in this direction.

*R1C10 Comment*: Figure 1. Could the resolution be improved? How important is the information on the summer/winter precipitation? It would be nice to see the RUSLE information on the map as well (maybe by using different colors for each point?) so the readers can see how rainfall erosivity is distributed within the domain.

*R1C10 Response*: Thanks for your remark. Figure 1 was modified as suggested. Information about the mean rainfall erosivity was added to the figure.

*R1C11 Comment*: Figure 3-6. Here too, the figure resolution needs to be improved.

*R1C11 Response*: Noted with thanks. Resolution of these figures was improved. Please note that the low-resolution was the result of copying figures from PDFs into the Word document. Hence, high-resolution PDF figures exist as well (and will be submitted for a possible typesetting stage).

*R1C12 Comment*: Figure 5-6. I have seen these figures several times... Maybe they can be presented as supplementary material instead of being included in the text?

*R1C12 Response*: Thanks for your remark. We would rather keep these figures in the main document since perhaps some of the soil erosion experts are not very familiar with the characteristics of the tested models. Hence, we decided to keep these figures in the main document.

*R1C9 Comment*: Line 419. The reference's year is incorrect - it should be 2017.

*R1C9 Response*: Indeed, the reference was not correct and was corrected. Thanks.

**Reviewer #2 (Assoc. Prof. Dr. Mark Silburn)**

*R2C0 Comment*: This a very well written, clear and concise paper. I have made only very minor edits to aid clarity. I have ranked Scientific significane as good only (rather than excellent) because the RUSLE is mature technicology.

*R2C0 Response*: We would like to thank Assoc. Prof. Dr. Mark Silburn for reviewing our manuscript. We very much appreciated the encouraging comments and overall positive evaluation on our study. Response to the specific suggestion is provided below. Thanks.

*R2C1 Comment*: Note town names in Figure 1 are unclear/hard to read.

*R2C1 Response*: This is a good suggestion. Thanks. Figure 1 was modified.

**Reviewer #3**

*R3C0 Comment*: The manuscript used 5-min and daily data from northern Germany to evaluate the accuracy of 4 different methods to estimate the rainfall erosivity (R) for sites where high temporal resolution rainfall data are not available. The manuscript is structured well, and mostly readable. The authors should be commended for being thorough and rigorous in their data analysis.

Presentation is ok, English expressions need attention to improve the quality of the manuscript. I have edited the abstract for the authors. Similar effort is probably needed throughout the manuscript, especially with respect to the tense. When we describe what we did, we need to use the past tense. Use the present when we cite other people's work, especially their observations and conclusions, because they have been published and, in a sense, made permanent.

*R3C0 Response*: We would like to thank Reviewer #3 for reviewing our manuscript. We very much appreciated the encouraging comments and overall positive evaluation on our study. As suggested we improved the English style of writing. However, we want to point out that the first author is a native speaker from Australia, which causes possible style differences (not referring to the tenses here). Response to the specific suggestion is provided below. Thanks.

**Specific comments**

*R3C1 Comment*: Fig. 1 – Poor quality. No longitude/latitude, no scale. So not all stations are located in Lower Saxony.

*R3C1 Response*: Noted with thanks. Figure 1 was modified according to the reviewer comments, the text has been modified.

*R3C2 Comment*: Fig. 3 – Use 'Computed R' for the first dashed box top left?

*R3C2 Response*: Although R is computed, it is still only an estimation of the 'true' value of R. We prefer to stick with 'Estimation of R' since there are more boxes with estimations of parameters, which are also (partly) computed. A differentiation here could lead to questions to the other boxes.

*R3C3 Comment*: Fig. 4 – Y-axis. If there is no scale, what is the point of having the unit?

*R3C3 Response*: Thanks for your remark. Figure 4 was modified.

*R3C4 Comment*: Ditto Fig. 5.

*R3C4 Response*: Noted with thanks. Changed as per figure 4.

*R3C5 Comment*: Line 38 - ha-1 h-1.

*R3C5 Response*: Corrected. Thanks.

*R3C6 Comment*: Line 57 – What do you mean by 'high frequency'?

*R3C6 Response*: Good point, thanks. Removed mention of 'high frequency'.

*R3C7 Comment*: Around line 60: To use the USLE/RUSLE, the following is needed: the R value, and monthly or half-monthly distribution of erosivity, and 10-year event EI30 values. While storm event characteristics are relevant, they are not needed, strictly speaking. Re-define and specify the research objectives. If event EI30 is included, some of the 4 methods are automatically disqualified. In other words, it is necessary to clearly define what aspect(s) of the rainfall erosivity to be estimated and the methods that can be used for that estimate.

*R3C7 Response*: Thanks for your remark. The description of study aims has been modified in order to make the steps conducted clearer for the reader. Hence, the main aim of this study was to focus on the annual rainfall erosivity since this is often used as input to the USLE-type models. Then we also investigated some detailed erosive events characteristics since this is often done in the studies that investigate erosive event characteristics.

*R3C8 Comment*: Around line 75: Fig. 1 indicates that stations outside of Lower Saxony were also included in the study. It is better to define the study area using latitude and longitude boundary to include Lower Saxony within that boundary.

10-year could normally be short for computing the R factor values. What is the interannual variability of annual precipitation and annual EI30 values among the recording stations with 5-min data?.

*R3C8 Response*: Noted with thanks. Additional explanation was added to the section 2 and one new reference was added as well. Below we have also added a figure showing monthly rainfall erosivity values for the Hannover station, please note that this interannual variability corresponds well to the study published by Ballabio et al. (2017).

[Figure]

Figure: Example of monthly rainfall erosivity values for Hannover station.

*R3C9 Comment*: Line 107: I understand that ',' is used for decimal places in many European countries. Not sure about the journal policy on this. I'd like to see the authors use 0.29, 0.72 etc as they appear in the original reference cited by the authors for an international readership.

*R3C9 Response*: Indeed, the typo was corrected. Thanks.

*R3C10 Comment*: Around Line 125: The authors include many covariables for EDK. May I suggest author include the mean annual precipitation (P) as one of the additional variables in Equation (5) because P is widely and reliably available and we the P and R are well related around the world listed below.

*R3C10 Response*: Thanks for the comment. Some additional discussion was added to the section 3.2.1. Please note that this study uses high-frequency data. Hence, we decided to use some precipitation characteristics that can be obtained directly from such data and not only from the mean annual precipitation, which can in some cases not be an optimal predictor of the mean annual rainfall erosivity.

*R3C11 Comment*: Around Line 155: How many parameters to be calibrated using observations for ARM? Do the parameters vary monthly or seasonally? How were parameter values estimated?

*R3C11 Response*: Noted with thanks. Additional explanation was added as suggested by the Reviewer #3.

*R3C12 Comment*: Line 18 – Linear transformation. How was this achieved? Did you have to extrapolate from 15-min and 7.5-min down to 5-min?

Again, how many parameters involved in the disaggregation method, how were they estimated, i.e. method of estimation?

*R3C12 Response*: Good point. We have added the following explanation in the manuscript: "More precisely, rainfall amounts of the $\Delta t = 7.5$ min level are distributed uniformly on 2.5 min time steps, which are subsequently aggregated to 5 min time steps " We also added details on the parameter estimation (data-driven, no optimization or calibration) in the text. However, we don't want to provide more details here since the focus is on the rainfall erosivity, not on the rainfall generators, which are published already. Also, the first reviewer suggested to remove figures from the rainfall generator description because these are 'well-known', so it seems impossible to satisfy both reviewers in this point. We extended the descriptions of the rainfall generators a bit, but prefer to show not more details.

*R3C13 Comment*: Around line 175: EDK was used to spatially interpolate the R factor. Why was this not used with covariables to interpolate daily rainfall amount before disaggregation?

*R3C13 Response*: Thanks for your remark. For the interpolation of daily rainfall amounts several of additional variables $Y_i$ would not have been available because they can only be derived from sub-daily resolution. Hence we applied the OK only.

*R3C14 Comment*: Line 194: '18 stations with the longest' record length: How long were they in number of years? Compute annual EI30 values to shed light on its underlying interannual variability to provide some empirical support for selecting 20% as a criterion to define a 'stable' estimate.

*R3C14 Response*: Thanks for your remark. 22 years was the average observation length. The below Figure shows an example of the variability in the annual rainfall erosivity for the Hannover station. Hence, additional explanation about the selection of the 20% threshold was added to the manuscript.

[Figure]

Figure: Example of variability in the annual rainfall erosivity values for the Hannover station. A longer data period was used for this station compared to the common period used in this study since this station has longer data availability.

*R3C15 Comment*: Result section: The result in relation to R is fine, but there is a need to justify the way erosive events were selected and aspects of these events were defined. Again, if the intent of the manuscript is to prepare the best possible input for the USLE/RUSLE, one needs to use events as defined in the USLE/RUSLE (Renard et al. 1997). If one selects events mostly for the sake of testing and comparing different rainfall interpolation approaches, it is useful to spell this out as one of the distinct research objectives.

*R3C15 Response*: Thanks for your comment. We have modified the description of research aims and added some additional discussion about this topic. Thanks.

*R3C16 Comment*: Conclusions: No. 1 Again, one questions whether we need to have the number of erosive events and event duration for the USLE/RUSLE. This depends on the research objectives. If we focus on event-level EI30 values, and other aspects, it is not even fair to include the Direct-R in the comparison.

No. 5 This needs to be assessed in the context of the underlying interannual variability of EI30 values for the region.

*R3C16 Response*: Good point. Conclusions section was modified in accordance to the Reviewers #3 comments. Moreover, please note that several changes were made in the previous parts of the manuscript.

*R3C16 Comment*: Modified abstract with track changes.

*R3C16 Response*: Thanks for your remark. Abstract was modified with most of the suggestions. Moreover, also other parts of the manuscript were modified in order to improve English style, especially tense.